# A divergent cyclic nucleotide binding protein promotes *Plasmodium* ookinete infection of the mosquito

Dominika Kwecka[1], Zhishuo Wang[2], Edvardas Eigminas[1], Lijia Liu[1], Jennifer C. Regan[1], Choel Kim[3], Nisha Philip[1]*

1 Institute of Immunology and Infection Research, University of Edinburgh, Edinburgh, United Kingdom,
2 Institute of Molecular Plant Sciences, School of Biological Sciences, University of Edinburgh, Edinburgh, United Kingdom, 3 Verna and Marrs McLean Department of Biochemistry and Molecular Pharmacology, Baylor College of Medicine, Houston, United States of America

* Nisha.Philip@ed.ac.uk

## Abstract

Colonisation of mosquitos by the malarial parasite is critically reliant on the invasive ookinete stage. Ookinete invasion of mosquito is coordinated by the apical complex, a specialised parasite structure containing components for secretion, attachment and penetration. While studies have investigated cytoskeletal and secretory elements, it is currently unknown if signalling modules are present or functional at the apical complex. Here we elucidate the role of a cryptic cyclic nucleotide-binding protein which we name CBP-O. PbCBP-O showed a marked localisation to the ookinete apex and disruption of the protein severely compromised ookinete invasion of mosquitos. Domain dissection analysis revealed that the N- and C-termini have distinct functions. Intriguingly, PbCBP-O exhibits dual binding specificity to both cGMP and cAMP. Our findings suggest the apical tip of the ookinete is a platform to transduce cyclic nucleotide signals essential for malaria parasite transmission.

## Author summary

Malaria parasites complete their life-cycle within mammalian and mosquito hosts requiring specialised forms to invade and establish infection in host tissue. In order to transmit to the mosquito, the parasite forms a motile ookinete that uses the apical end to attach and penetrate the mosquito midgut membrane. Previous studies have identified several proteins localised to the apex of the ookinete. However, the functions of many of these proteins are unknown. Here, we dissect the role of a putative cyclic nucleotide-binding protein, CBP-O, in establishing ookinete infection within the mosquito. We discovered that in the absence of the protein, efficiency of transmission is significantly compromised. Interestingly the ability of ookinetes to move was not affected, but the parasites were unable to

**Data availability statement:** All data are in the manuscript and/or supporting information files (ST2).

**Funding:** This work was supported by University of Edinburgh startup funds and Wellcome Trust Seed Award (212398/Z/18/Z) to N.P and the UKRI Biotechnology and Biological Sciences Research Council (BBSRC: BB/M010996/1) to D.K. Wellcome Trust:https://wellcome.org/ UKRI-BBSRC: https://www.ukri.org/councils/bbsrc/ Funders did not play any role in the study design, data collection and analysis, decision to publish, or preparation of the manuscript.

**Competing interests:** The authors have declared that no competing interests exist.

pass through the midgut membrane. We also reveal that both the N- and C- terminal domains of the protein perform distinct functions and are both required for parasite transmission. Sequence and spatial conservation of the protein across the phylum Apicomplexa suggests an equivalent and crucial function of CBP-O in these parasites.

## Introduction

Malaria continues to be a major cause of childhood disease resulting in 263 million cases and almost 600,000 deaths in 2023 [1]. The Apicomplexan parasite, *Plasmodium*, responsible for causing malaria, completes its complex lifecycle in a vertebrate host where it causes disease and, a mosquito vector that is necessary for parasite transmission. Transmission is initiated when the mosquito takes a blood meal containing *Plasmodium* male and female gametocytes. Upon encountering the mosquito environment gametocytes are activated and develop into gametes that fertilise to form a zygote. Within 16–24 hours zygotes transform into polarised, crescent-shaped ookinetes. Ookinetes are specialised for seeking, interacting with and penetrating the insect midgut epithelium for successful colonization of the mosquito.

Cyclic nucleotide signalling mediated by guanosine 3',-5'-cyclic monophosphate (cGMP) and adenosine 3',-5'-cyclic monophosphate (cAMP) is essential for maintaining the invasive life style of *Plasmodium* [2,3]. In ookinetes, coordination of cGMP synthesis by guanylate cyclase β and hydrolysis by phosphodiesterase PDEδ is critical for timely regulation of cGMP effector kinase Protein Kinase G (PKG) and governs ookinete morphology and motility [4,5]. GCβ mutant ookinetes show a strong motility defect which can be recapitulated by pharmacological inhibition of PKG demonstrating a critical function for cGMP mediated signalling cascade in ookinete gliding [6]. While gliding motility is essential to power ookinete displacement, signalling processes regulating ookinete passage into mosquito tissue are poorly understood.

Only three cellular effectors of cyclic nucleotide signalling are predicted in the *Plasmodium* genome which include PKG, cAMP-dependent protein kinase (PKA) and a cyclic nucleotide domain containing protein, designated as exchange protein activated by cAMP or EPAC [2]. While EPAC is not essential for the *P. falciparum* asexual blood stages and accumulates mutations in culture adapted parasites, the gene locus is preserved in field isolates and rodent parasite species [7–9]. This suggests EPAC is either required for *in vivo* growth of asexual stages or is important for parasite transmission. A large-scale screen of the molecular architecture of the apicomplexan apical complex identified *P. berghei* homologue of PfEPAC, PBANKA_1025300 localised to a specific apex compartment in all invasive or zoite stages of the parasite: merozoites, ookinetes and sporozoites [10]. Zoites of the Apicomplexa phylum are defined by a specialised structure at their polar tip called the apical complex [11]. The apical complex is not only important for maintaining the cytoskeletal structure of the zoite, it equips the parasite with a channel to secrete molecules that facilitate motility, as well as attachment and invasion of the host cell [12–15].

The distinct concentration of PBANKA_1025300 to the apex of the ookinete implies a critical function at the apical complex; however, it is unknown whether PBANKA_1025300 binds cAMP or coordinates ookinete motility and invasion. In this study, we systematically dissected the role of putative EPAC in ookinete function and discovered the protein regulates efficiency of ookinete invasion of the mosquito midgut epithelium. We reveal distinct functions of the N-terminal and C-terminal domains and provide the first direct evidence that the protein exhibits dual-specificity to both cAMP and cGMP. Finally, we found no evidence the protein is a guanine exchange factor and therefore renamed EPAC to *cyclic nucleotide binding protein in ookinetes*, or CBP-O.

## Results

### CBP-O is localised at the tip of ookinetes and enables mosquito infection

We first interrogated if PBANKA_1025300 is the predicted rap guanine nucleotide exchange factor [16]. Mammalian isoforms of EPAC are divided into a regulatory and a catalytic domain (Fig 1A). The regulatory domain contains the cyclic nucleotide binding domain or CNB (two CNBs in the case of EPAC2) and a Dishevelled, Egl-10/pleckstrin (DEP) domain. The catalytic portion is formed by the domains involved in the guanine nucleotide exchange, namely Ras exchange motif (REM) domain, RAS association (RA) domain, and CDC25-homology domain (CDC25HD) [17,18]. In contrast, the domain architecture of the putative CBP-O in *Pb* or *Pf* does not support function as a guanine exchange factor (Fig 1A). Consistent with findings noted for PF3D7_1417400 by Patel, Perrin and colleagues, we found the N-terminus of PBANKA_1025300 contains no recognizable domains and, structures required for the recruitment of Rap1 GTPase are absent. However, the C-terminus contains four predicted cyclic nucleotide binding domains (CNBs), out of which CNB1–3 are consecutive and the final CNB4 caps the protein. Additionally, a fifth highly degenerate domain, CNBX is inserted

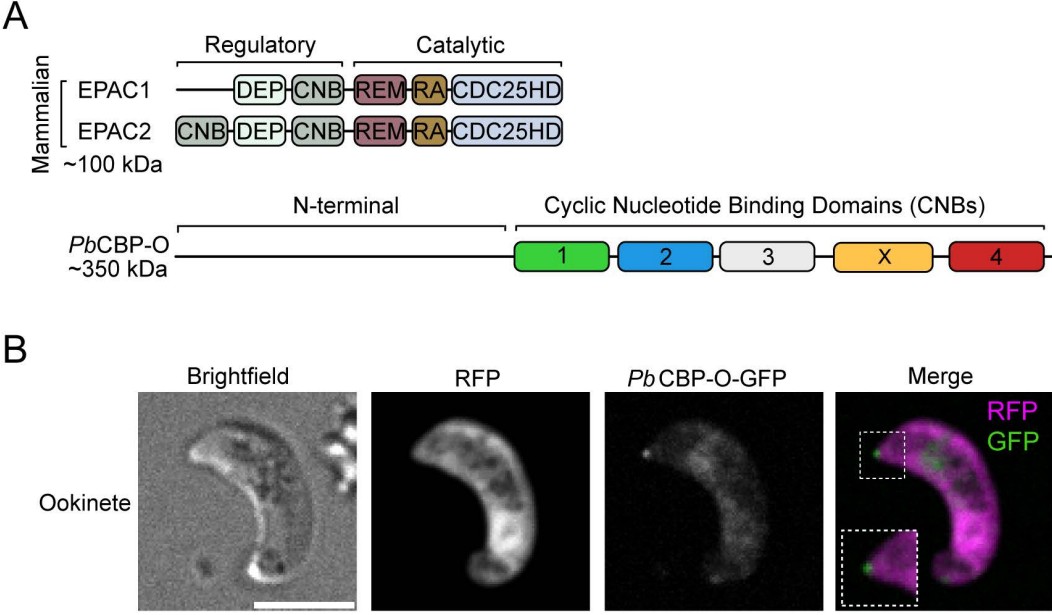

**Fig 1. Domain architecture and cellular localization of PbCBP-O. (A)** Schematic representation of the domains of mammalian EPAC isoforms composed of a regulatory and a catalytic portion. PbCBP-O shows an uncharacterised N-terminal domain, followed by five putative cyclic nucleotide binding domains (1-4 and X). DEP = Dishevelled, Egl-10/pleckstrin, cyclic nucleotide binding domain = CNB, Ras exchange motif = REM, RAS association = RA and CDC25-homology domain = CDC25HD. **(B)** Live imaging of GFP-tagged PbCBP-O shows expression at the ookinete's apical tip. Scale bar = 5µm.

between CNB3 and 4. This suggests PBANKA_1025300 might be a novel cyclic nucleotide responsive protein which has acquired functions specific to parasitic life.

Tagging PbCBP-O by adding GFP to the C-terminus revealed the protein is predominantly expressed as a distinct punctum at the apical tip of ookinetes, with some diffuse staining in the cell body as well (Figs 1B and S1A). This is consistent with previous studies showing the localization of PbCBP-O and its Toxoplasma orthologue (TGME49_219070) at the apex of the conoid [10]. The appearance of a small variation in positioning of CBP-O in our study compared to ookinetes from previous findings might be attributed to use of the ookinete plasma membrane marker P28 in the Koreny study [10]. Alternatively, spinning disc confocal imaging on live ookinete cells without labelling in our study, potentially captured apical protrusion of ookinetes. Interestingly, localization of CBP-O in our study is akin to the *T. gondi* orthologue in the extruded conoid [10]. Nevertheless, both studies confirm the expression of CBP-O at the Apicomplexan conoid. The conoid is located within the apical complex, a specialised structure containing the machinery required for invasion such as secretory organelles and specialised cytoskeletal elements [19].The highly specific apical localization of PbCBP-O in zoites implies a role during parasite invasion.

To determine the function of PbCBP-O in zoites, we deleted the complete ORF using CRISPR-Cas9 and replaced the locus by a GFP cassette constitutively expressing across the whole life cycle (Fig 2A). Diagnostic PCR confirmed successful integration of the repair fragment at the *cbp-o* locus (S1B Fig). Successful deletion of *pbcbp-o* indicated that it is not essential for asexual stage development and when compared to wild-type, PbΔCBP-O parasites grew at a similar

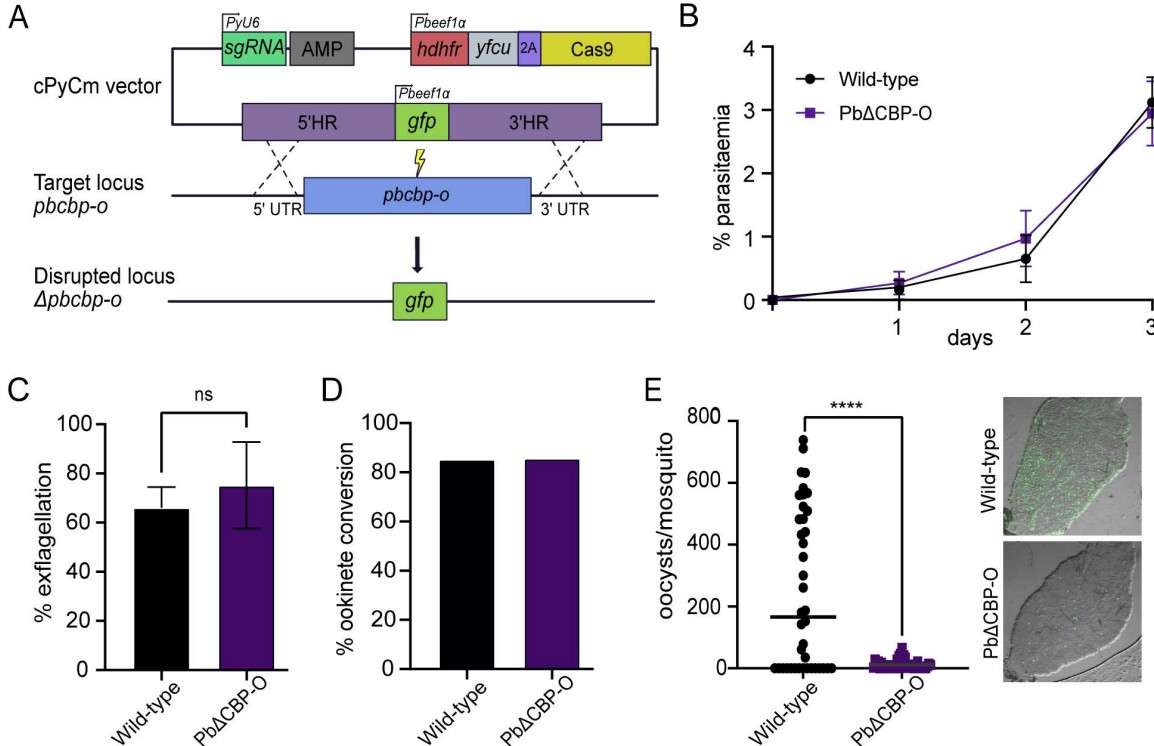

**Fig 2. PbCBP-O regulates parasite transmission into the mosquito. (A)** Schematic of strategy used for deletion of *pbcbp-o*. **(B)** In asexual blood stages, both wild-type and PbΔCBP-O parasites grow at a similar rate. Parasitaemia over three replication cycles. Error bars, ±SD and n = 3 independent experiments. **(C)** Exflagellation is not affected in PbΔCBP-O parasites (n = 2 independent experiments, paired t-test). **(D)** Ookinetes are formed in the absence of PbΔCBP-O parasites (n = 1). **(E)** PbΔCBP-O parasites fail to infect the mosquito midgut. Images show a mosquito midgut and infected green oocysts (dissections performed at day 7 post infection, wild-type n = 40, PbΔCBP-O n = 43, unpaired t-test). In all relevant panels, statistical significance is indicated as: ns = not significant, and **** p < 0.0001. For panels B, C, D and E, medians are indicated.

rate over three replication cycles (Fig 2B). PbΔCBP-O parasites exhibit no change in parasitaemia and form gametocytes in mice, and therefore it is unlikely that PbCBP-O has a crucial function during the asexual stage. These observations concur with previous studies that showed *P. falciparum* CBP-O is dispensable for asexual growth *in vitro* [7,9]. To examine the role during transmission, PbΔCBP-O gametocytes were obtained from sulfadiazine treated mice and investigated for their ability to activate in response to xanthurenic acid in ookinete media. No defects in exflagellation were observed (Fig 2C) and 24 hours later when gametocyte conversion into ookinetes was assessed, no defect was observed in the ability to form ookinetes (Fig 2D). However, when *Anopheles stephensi* mosquitoes were fed on mice infected with wild-type or PbΔCBP-O parasites and midguts dissected 7 days later to check for oocyst presence, we observed a striking 98% decrease in oocyst load (Fig 2E).

## CBP-O deficient ookinetes are motile but cannot cross the midgut epithelium

Gliding motility is critical for midgut traversal of ookinetes [20,21]. We assessed the speed of PbΔCBP-O ookinetes using an *in vitro* Matrigel-based assay. The ookinetes displayed a 3- D meandering motion and, speed of both wild type and PbΔCBP-O ookinetes were comparable (Fig 3A; S1 and S2 Movies). Moreover, over 80% of mutant and wildtype ookinetes showed the characteristic left-handed chiral movement (S3 Fig; S3 and S4 Movies). To determine if the defect in oocyst production was due to deficient ookinete development *in vivo*, mosquitoes were fed on mice infected with PbΔCBP-O parasites and 19 hours after transmission, mosquito midguts were dissected to assess ookinete presence. No difference in ookinete numbers was observed between wild-type and PbΔCBP-O parasites (Fig 3B) confirming that PbΔCBP-O can produce mature ookinetes *in vivo.* Secretion of adhesins and proteases from micronemal organelles through the apical complex facilitates ookinete motility, escape from the blood bolus and breaching of the midgut epithelium [22–25]. However, total levels of micronemal proteins such as CTRP, WARP and Chitinase in PbΔCBP-O remained comparable to wild type ookinetes (Fig 3C). Nor did we observe any differences in chitinase secretion.

Ookinetes penetrate the midgut epithelium and, passage into the lumen between the midgut epithelium and basal lamina layer is obligatory to establish a mosquito infection. To examine if PbΔCBP-O crossed the midgut epithelial layer, infected midguts were dissected 24–25 hours after transmission followed by an immunofluorescence assay. The blood bolus was removed, midguts washed in PBS and, P25 antibody staining was used to count the parasites that had crossed the epithelium. Strikingly, removal of CBP-O resulted in a significant decrease in ookinete numbers in the midgut epithelium and, consequently there were fewer sporozoites in the salivary glands (Fig 3D & 3E). To understand if the decreased infection in midguts is due to failure of ookinetes to invade the midgut epithelium or parasite elimination by the mosquito immune system, we analysed expression of *An. stephensi* SRPN6 immune gene. AsSRPN6 is a protease inhibitor that is transcriptionally activated in response to ookinete invasion [26]. Compared to uninfected blood-fed mosquitoes, midguts containing wildtype parasites had significantly elevated AsSRNP6 mRNA expression (Fig 3F). Notably, PbΔCBP-O failed to induce levels of SRNP6 compared to wildtype parasites indicating that the mosquito immune system doesn't perceive the mutant ookinetes.

To test whether bypassing the midgut barrier would rescue the mutant phenotype and allow PbΔCBP-O to mature into salivary gland sporozoites, we generated mutant and wildtype ookinetes *in vitro* and injected 700 ookinetes into the mosquito haemocoel. When salivary gland sporozoites were assessed 19 dpi, comparable numbers of sporozoites were obtained from wildtype and PbΔCBP-O ookinete injections (Fig 3G).

Taken together, our experiments reveal that ookinetes require CBP-O to invade the mosquito midgut and the protein may regulate efficient traversal through midgut epithelium.

## The CNB and N-terminal domains are required for efficient midgut invasion

PbCBP-O contains an N-terminal region without any recognisable domains, but five predicted cyclic nucleotide binding domains (CNB) span the second half of the protein (Fig 4A). To investigate the contributions of the N- and C- regions,

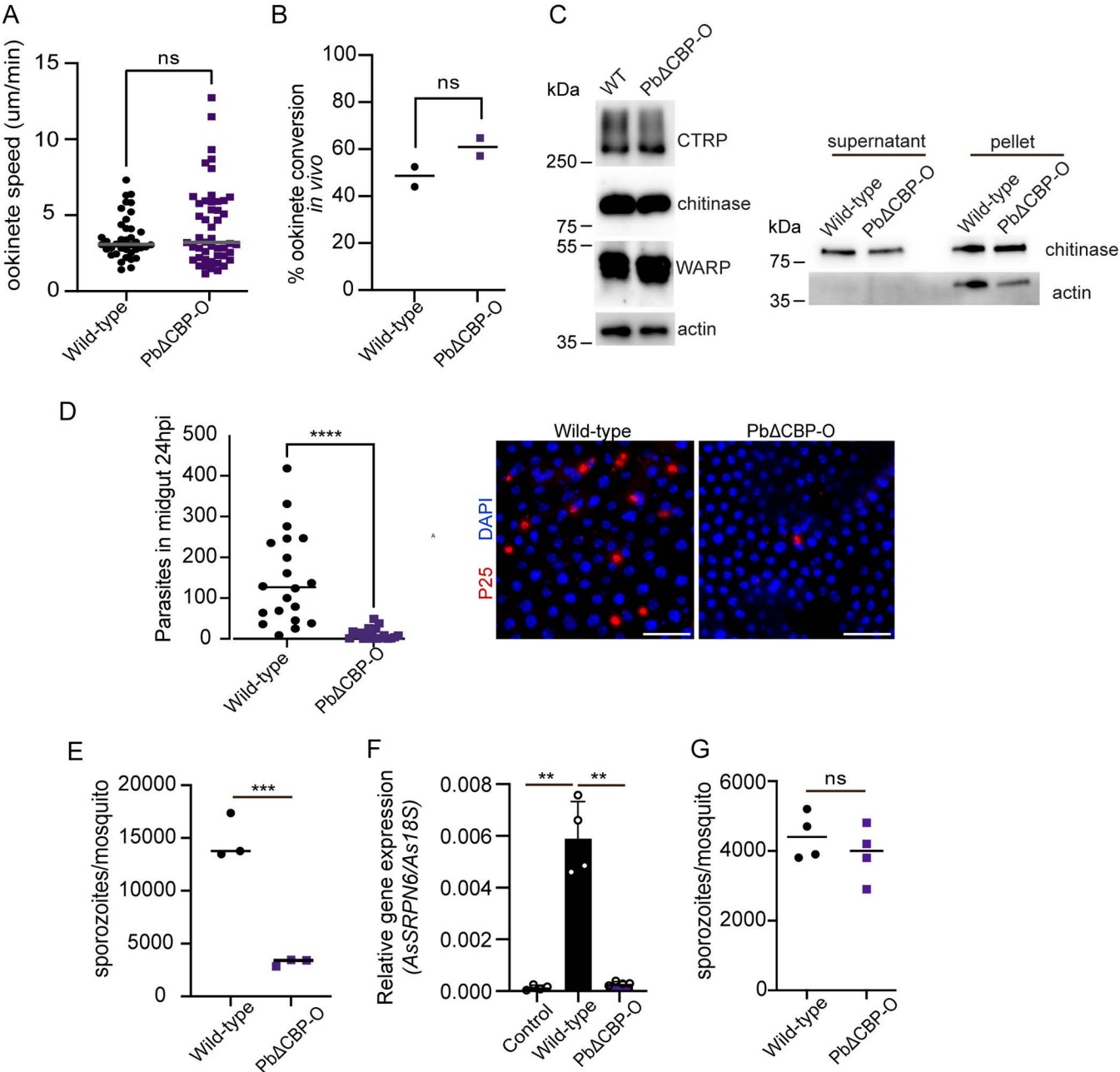

**Fig 3. PbCBP-O is required for efficient midgut traversal. (A)** PbΔCBP-O ookinetes show regular motility. Average speed is 3.07 and 3.2 µm/min for wild-type and ΔCBP-O, respectively (wild-type n = 38; PbΔCBP-O n = 48, representative data from 3 independent experiments, unpaired t-test). **(B)** Comparable ookinete maturation in the mosquito midgut is observed in PbΔCBP-O and wildtype parasites. Nineteen or twenty-one hours post-infection, six midguts were scored per replicate (n = 2). Significance was analysed using an unpaired t-test. **(C)** Expression of micronemal proteins including CTRP, WARP and chitinase are unaffected in the absence of CBP-O. Actin is used as loading control (left panel). Chitinase secretion is maintained in PbΔCBP-O ookinetes. Western blot of culture supernatants from wild type and PbΔCBP-O ookinetes show similar levels of chitinase secretion (right panel), representative data from 2 independent experiments, **(D)** Immunofluorescence showing decreased ookinete presence in the basal side of mosquito midguts. Midguts were fixed and stained with P25 (plasma membrane marker) and DAPI 24-25 hours post-infection. P25⁺ parasites were quantified, and single dots show number of ookinetes counted per midgut (n = 20 midguts per genotype, unpaired t-test.). Scale bar = 200µm. **(E)** Quantification of salivary gland sporozoites. Each data point is an average of five infected mosquitoes (representative data from two independent experiments). **(F)** Expression of *An. stephensi SRPN6* mRNA relative to *An. stephensi rps7* in mosquitoes that were fed uninfected blood (control), fed with wt or PbΔCBP-O infected blood (Data are mean ± SD, n = 25 midguts per condition, pooled from 4 independent experiments, unpaired t-test with Welch's

correction). **(G)** Quantification of sporozoites isolated from salivary glands nineteen days post ookinete injections. Each data point is an average of five infected mosquitoes (representative data from n = 2 independent experiments, unpaired t-test). In all relevant panels, statistical significance is indicated as: ns = not significant, ** = p < 0.01, *** = p < 0.001 and **** = p < 0.0001.

three domain deletion mutants tagged with GFP were generated using CRISPR-Cas9 in a parasite line (1804cl1) that constitutively expressed red fluorescence protein [27]. The mutants include: (i) removal of the distal and most conserved CNB, ΔCNB4: 2940–3136 a. a. deleted, (ii) all five CNBs removed ΔCNB1–4/X:1716–3136 a. a. deleted and (iii) the N-terminus excision ΔN-term: 1–1694 a. a. deleted. Similar to PbΔCBP-O, the domain mutants displayed wild-type rates of ookinete transformation (Fig 4B). However, both N-terminal deletion or complete removal of CNBs decreased oocyst load intensity by 96.63% and 96.5%, respectively, signifying the crucial role of the domains in establishing a midgut infection (Fig 4C). The individual CNB4 mutant exhibited no defects in oocyst load (Fig 4C), suggesting that the remaining CNBs are sufficient to maintain CBP-O function. Interestingly, while the infection prevalence of all mutant lines is comparable to the wildtype, the intensity of infection is significantly reduced in ΔCBP-O, ΔCNB1–4/X and ΔN-term mutants (Fig 4C) suggesting that CBP-O ensures efficient invasion of the midgut. When we tested for the ability of parasites to penetrate the midgut epithelium, significant proportion of ΔCNB1–4/X and ΔN-term ookinetes failed to cross the epithelial barrier (Fig 4D and 4E). Accordingly, reduced midgut invasion and oocyst intensity impacted sporozoite load in salivary glands (Fig 4F). These findings demonstrate that both the N-terminal region and CNB domains are important for CBP-O to regulate ookinete passage into mosquito tissue.

Given the intriguing localisation of CBP-O to the extreme apex of the ookinete, we asked whether the domain mutants could reach the ookinete tip. Live microscopy of ΔCNB1–4/X and immunofluorescence imaging of ΔCNB4 revealed that in the absence of the CNBs, localization of the protein to the apical tip of the ookinete is not disrupted (Figs 5 and S4A). However, upon removal of the N-terminal domain (ΔN-term), the truncated protein remained within the cell body and was no longer targeted to the tip (Fig 5). Immunofluorescence imaging of merozoites in mature schizonts revealed similar localisation features (S4B Fig). While we are not aware of a consensus sequence or structure determining protein trafficking to the apical tip, our results reveal that the N-terminal domain secures CBP-O localisation to the apex in both merozoites and ookinetes.

## CNB domains of CBP-O reveal structural and sequence variation

Recognition of the secondary messengers, cGMP and cAMP, by cyclic nucleotide binding domains can transduce both internal and external signals to regulate diverse cellular responses [28]. Functional CNBs are commonly composed of a variable number of α-helices, the three major ones being αA, αB and αC, and an eight strand β-barrel structure that functions as a basket. The cyclic nucleotide binds at the base of the β-barrel protected from phosphodiesterases while the αC helix caps the bound cyclic nucleotide to increase binding affinity. The Phosphate Binding Cassette (PBC), the key structural motif responsible for binding to the cyclic nucleotide, comprises a short helical turn or P-helix and a loop between β6 and β7 [28].

To investigate whether the CNBs of PbCBP-O adopt the predicted architecture, their primary sequences and AlphaFold models (Fig 6A and 6B) were compared to the crystal structures of the most carboxy-terminal CNB domain (PfCNB-D) of *P. falciparum* cGMP-dependent protein kinase (PfPKG) and the CNB domains of *P. falciparum* cAMP dependent kinase regulatory subunit (PfPKA-R) [29,30]. Initial analysis suggests that the CNBs of PbCBP-O have an eight β-barrel structure with flanking helices as in PfCNB-D of PfPKG and PfCNB2 of the regulatory domain in PfPKA, with the PBC located inside the β-barrel and the capping αC helix (Fig 6B). However, the sequence alignment revealed a few key differences (Fig 6A). CNB3 shows a large insertion between β4 and β5, and CNBs 1–3 showed extra helices close to αC. Moreover, sequence analysis of CNB2 indicated that it lacks key PBC residues required for cGMP binding. Previous co-crystal structures of CNB domains from PfPKG bound with cGMP revealed a conserved glutamate and arginine within the PBC

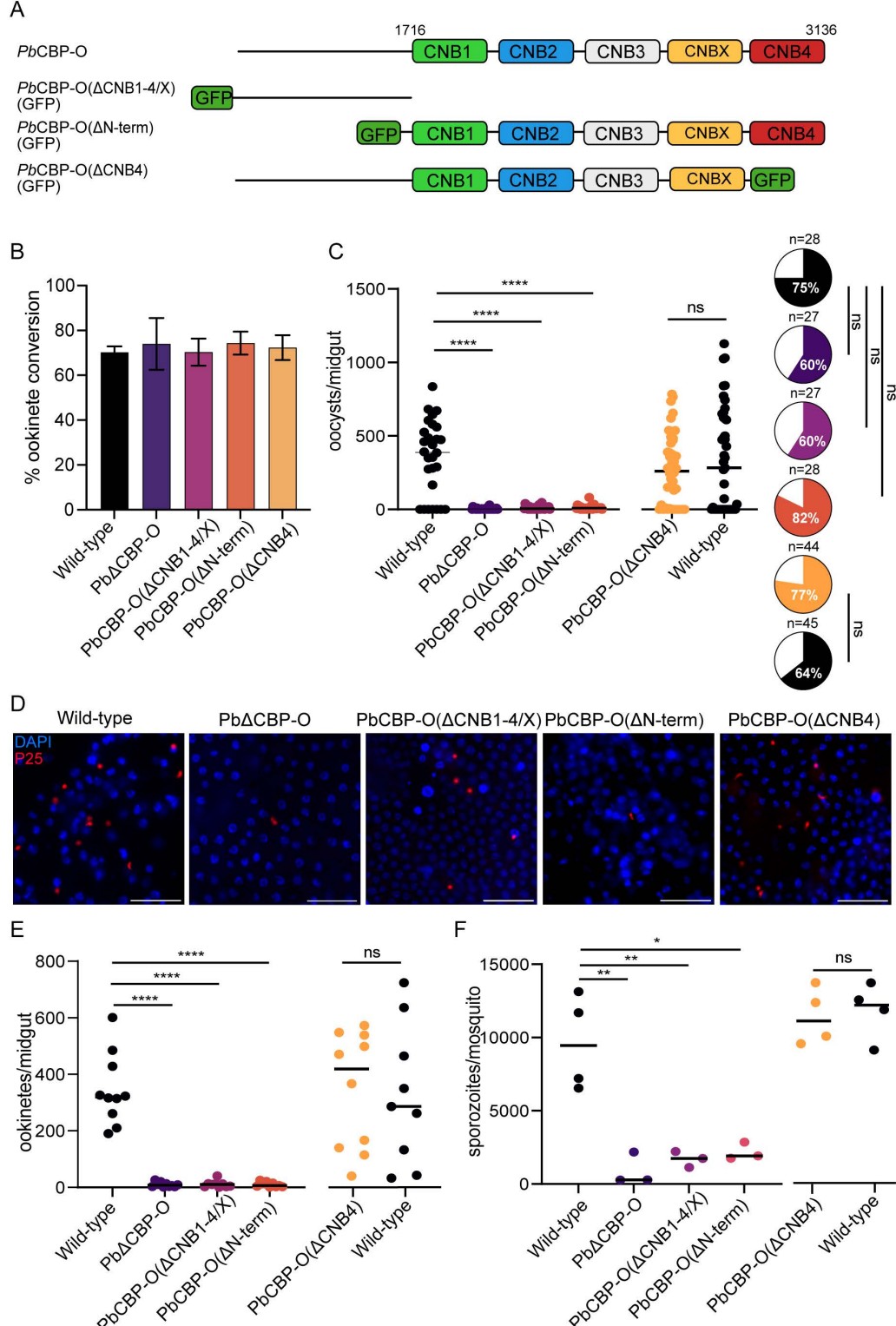

**Fig 4. N- and C- terminal domains of PbCBP-O are important for ookinete function. (A)** Schematic representation of full length and GFP tagged mutant PbCBP-O lacking N-terminus (ΔN-term) or cyclic nucleotide binding domains (ΔCNB). All mutants were generated in a background line

constitutively expressing RFP. **(B)** Ookinete maturation of full length ΔCBP-O and the individual domain mutants (ΔN-term, ΔCNB1-4/X and ΔCNB4) are comparable to wildtype. n = 3 independent experiments. **(C)** Oocyst numbers in midguts infected with ΔCBP-O (null), ΔN-term, ΔCNB1-4/X, ΔCNB4 and wild-type 12 days post-infection. A dot represents the number of oocysts in a single midgut (n = number of midguts dissected, unpaired t-test). Pie chart shows infection prevalence (Fischer's exact test). **(D)** Immunofluorescence imaging of ookinetes and early oocysts in mosquito midguts infected with ΔCBP-O (null), ΔN-term, ΔCNB1-4/X, ΔCNB4 and wild-type 24 hours post-infection. Scale bar = 200µM. **(E)** Quantification of ookinetes/early oocysts in the midgut. n = 10 midguts per genotype. **(F)** Quantification of salivary gland sporozoites. Each data point is an average of five infected mosquitoes. In all relevant panels, statistical significance is indicated as: ns = not significant, * p = 0.05, ** = p < 0.01, *** = p < 0.001 and **** = p < 0.0001. Experiments in C, D, E and F were independently repeated twice, and data is shown from one representative experiment.

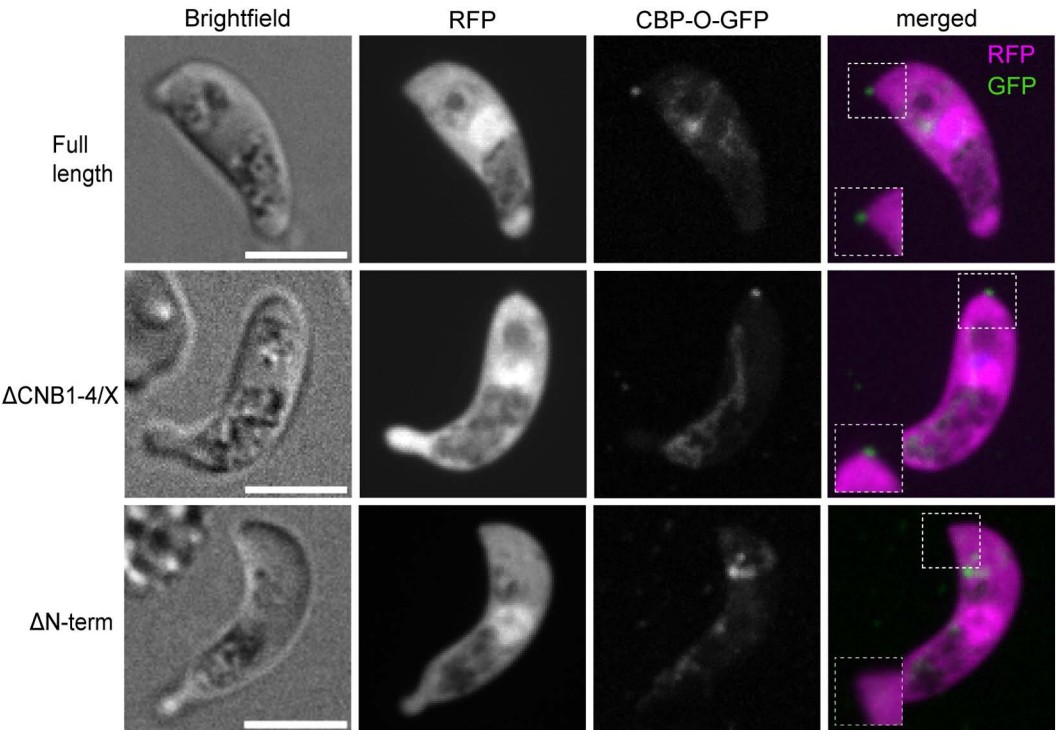

**Fig 5. The N-terminal domain guarantees accurate protein localisation.** Live fluorescence microscopy with GFP-tagged CBP-O reveals requirement of the N-terminus for targeting CBP-O to the ookinete apex. The full-length and CNB domain mutant protein (ΔCNB1-4/X) maintain apical localisation. Scale bar = 5µM.

that mediate binding to the cyclic nucleotide [31]. These residues are conserved in CNB1, 3 and 4, but in CNB2 they have been replaced with a histidine (His2046) and a serine (Ser2051). Mammalian and *Plasmodium* PKGs also present a highly conserved threonine (*Pf*CNB-D Thr494) following the arginine (*Pf*CNB-D Arg493) in the PBC, whereas the *Pf*PKA-R-CNB domains contain an alanine at the corresponding position. The Thr494 in *Pf*PKG mediates high affinity interaction to the cyclic nucleotide, and when mutated there is a dramatic decrease in PKG activity [32,33]. CNB1, 3 and 4 of PbCBP-O present a threonine (Thr1813, Thr2237 and Thr3024, respectively) following the highly conserved arginine, suggesting similarity to *Pf*CNB-D and that they may bind cGMP. On the other hand, CNB2 shows an isoleucine (Ile2056) at the corresponding position. Taken together, these residue differences suggest that CNB2 cannot bind cyclic nucleotides. Another determining feature of CNBs is a capping interaction to enclose the cyclic nucleotide in a hydrophobic pocket and shield it from solvent. In *Pf*CNB-D, capping is mediated by Arg484 in the PBC and Gln532/Asp533 in the α-C helix [29]. In *Pf*PKA-R the capping is mediated by a tyrosine residue (Tyr424) also located in the αC helix [30]. In PbCBP-O, the

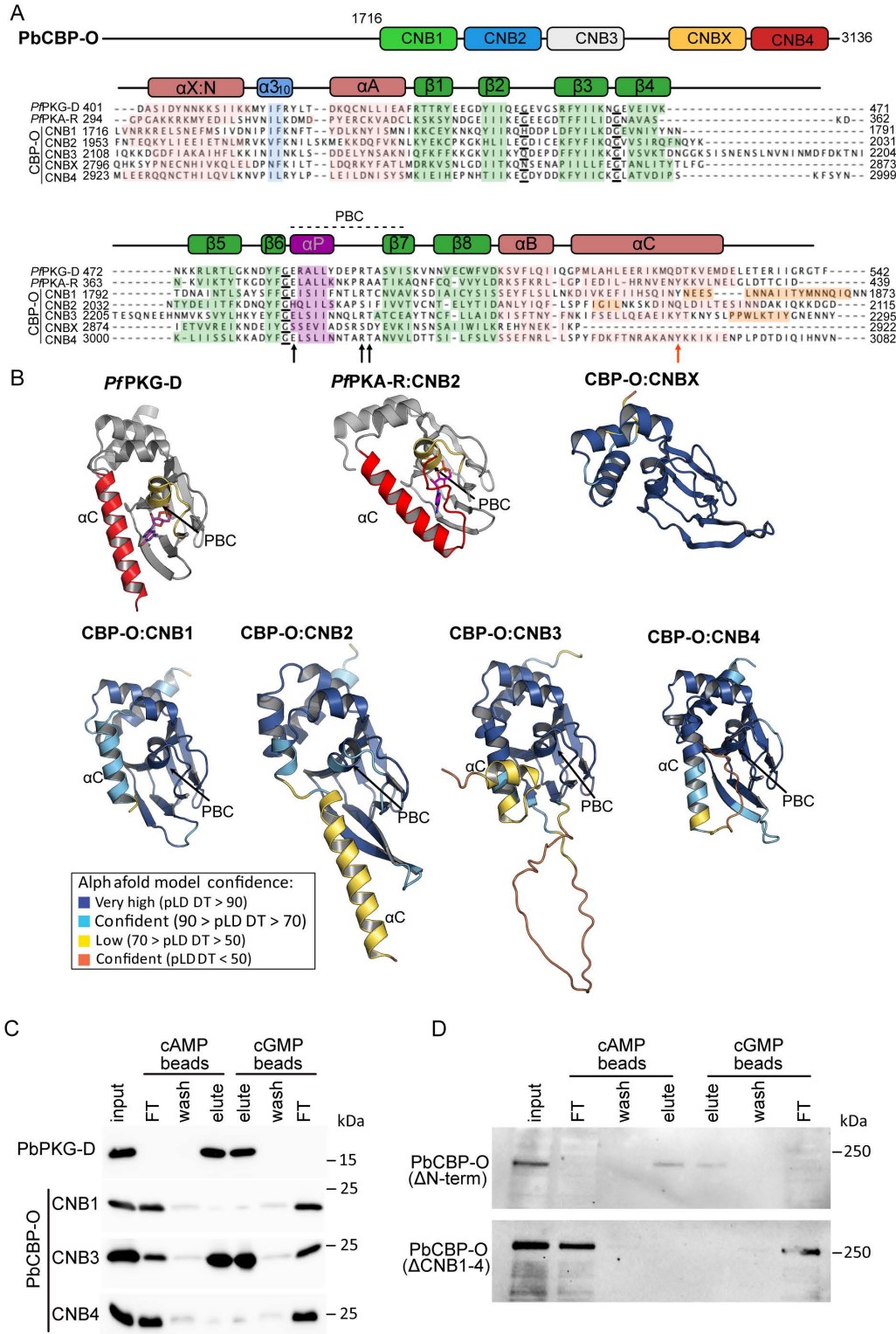

**Fig 6. Structural conservation and binding properties of putative CNB domains.** (A) Sequence alignment of the five CNBs of PbCBP-O with CNB-D of PfPKG and CNB2 of PfPKA-R illustrates that the domain organization is conserved. α-helices are shown in peach, β-barrels in green and the

phosphate binding cleft (PBC) in magenta. Black arrows refer to the conserved glutamate (E), arginine (R) and threonine (T) in the PBC of *Pf*CNB-D ($E_{484}$, $R_{493}$ and $T_{494}$) and the PbCBP-O CNB1 ($E_{1803}$, $R_{1812}$, $T_{1813}$), CNB3 ($E_{2227}$, $R_{2235}$, $T_{2237}$) and CNB4 ($E_{3014}$, $R_{3023}$, $T_{3024}$). The orange arrow marks the Tyrosine (Y) capping residue for PbCBP-O CNB1 ($Y_{1853}$), CNB3 ($Y_{2275}$) and CNB4 ($Y_{3062}$) which is equivalent to the Tyrosine capping residue for *Pf*PKA-R ($Y_{424}$). Underlined residues highlight the conserved glycine triad. (B) Comparison of Alphafold model predictions of PbCBP-O-CNB1–4 and X to the crystal structure of *Pf*CNB-D bound with cGMP (PDB ID: 4OFG) and CNB2 of *Pf*PKA-R bound with cAMP (PDB ID: 5KBF). The PBC and αC helix of *Pf*CNB-D/*Pf*PKA-R -CNB2 are coloured in yellow and red respectively. The bound cGMP and cAMP are shown in sticks. A disordered insertion is predicted between $β_5$ and $β_6$ barrels of CNB3. Images were created in PyMOL. (C) cAMP and cGMP pull down assay with recombinantly expressed 6XHis tagged CNB-D from PbPKG and CNB1, 3 and 4 of PbCBP-O revealing that CNB3 binds both cAMP and cGMP. (D) Lysate from mature schizonts expressing GFP tagged ΔCNB1–4/X and ΔN-term isoforms of CBP-O were incubated with cAMP or cGMP agarose. Western blot analysis (α-GFP) revealed that upon removal of the N-terminal, the protein retains the capacity to bind cyclic nucleotides, while in the absence of CNB1–4/X, the mutant protein is unable to associate with cAMP or cGMP. n = 4 and 3 independent replicates for (C) and (D) respectively, and data is shown from a single representative experiment.

capping mechanism in CNB1,3 and 4 might be similar to *Pf*PKA-R, due to an equivalently positioned tyrosine (Tyr1853, Tyr2275 and Tyr3062, respectively) in the αC helix. PbCBP-O includes a degenerate CNB (CNBX, located between CNB3 and CNB4) with a lower degree of conservation than the other four CNBs. While the Alphafold model shows an eight β-barrel structure with flanking helices as in other CNBs (Fig 6B), this domain lacks all key residues for binding cyclic nucleotide. From our structure-based analysis, we infer that CNB1, 3 and 4 of CBP-O contain a blend of features from CNB domains of *Pf*PKA and *Pf*PKG and could support cyclic nucleotide binding.

### CNBs exhibit binding to both cAMP and cGMP

The sequence alignment predicts that PbCBP-O contains three functional CNB domains. However, as the domains displayed features characteristic of both PKA and PKG, cyclic nucleotide specificity could not be confidently predicted by structural modelling. To examine their binding capabilities, 6X his-tagged CNB1, CNB3 and CNB4 were recombinantly expressed in *E. coli* and CNB-D from PbPKG was used as control. PbCNB2 and PbCNBX was excluded from our analysis since they presented features of a degenerate CNB domain incapable of cyclic nucleotide binding. Recombinant PbPKG-D, PbCNB1 and PbCNB3 were easily recovered from the soluble fraction, but PbCNB4 aggregated into inclusion bodies (S5 Fig). To recover PbCNB4, inclusion bodies were lysed in 8M urea followed by buffer exchange through dialysis to reconstitute the protein in PBS.

To analyse cyclic nucleotide binding properties, recombinant protein forms of PbPKG-D, PbCBP-O's CNB 1,3 and 4 were incubated with agarose beads linked to either cAMP or cGMP. Cyclic nucleotide bound proteins were recovered by heating with 1X SDS loading buffer and analysed by western blotting. As expected PbPKG-D showed strong binding to cGMP, but also displayed affinity to cAMP beads. This is not surprising since *Pf*PKG-D is known to demonstrate low affinity binding to cAMP with a $K_d = 7.4 \mu M$ [29] and cAMP ligand density of the affinity matrix is 6mM. While CNB1 and CNB4 from CBP-O lacked the ability to bind either cAMP or cGMP, CNB3 showed affinity for both cAMP and cGMP (Fig 6C). The results obtained for PbCNB4 were surprising as the domain is predicted to have higher conformational conservation compared to CNB1, 2 and 3. However, purification of PbCNB4 from inclusion bodies might have failed to reconstitute the correct folding, and therefore lacked binding. Cyclic nucleotide binding capacity of PbCNB3 is intriguing and demonstrates that the large insertion between β4 and β5 does not prevent binding to either cAMP or cGMP. Finally, to investigate whether the CNB domains are functional *in vivo*, we took advantage of the ΔCNB1–4/X and ΔN-term mutant lines and utilised schizonts containing mature merozoites that permitted larger preparative cultures and greater number of cells expressing CBP-O compared to ookinetes (~14n vs 1n). Protein lysates from schizont parasites expressing ΔCNB1–4/X and ΔN-term isoforms were incubated with cAMP or cGMP agarose and examined for binding capacity. While CNB1–4/X removal prevented protein binding to cAMP and cGMP, the N-term mutant retained cyclic nucleotide binding activity (Fig 6D).

Taken together these results demonstrate that the C-terminus of PbCBP-O is indeed a cyclic nucleotide binding domain and at least one of the predicted CNBs, CNB3 can directly interact with both cAMP and cGMP.

## Discussion

PKA and PKG are the only well-characterised cyclic nucleotide effectors in *Plasmodium* and show high specificity to cAMP and cGMP respectively [2,3,34]. However, a third putative cyclic nucleotide binding protein is encoded in the *Plasmodium* genome, whose function remained elusive. In this study, we show that CBP-O is critical for *P. berghei* ookinetes to invade the mosquito midgut and transmit into the vector.

Domain analysis of PbCBP-O revealed the presence of four putative cyclic nucleotide binding (CNB) domains at the C-terminus of the protein; it also highlighted the absence of recognisable guanine nucleotide exchange factors. PbCBP-O lacks the domain necessary to catalyse the exchange of inactive GDP for GTP, nor does the protein contain the RA domains necessary for binding to RAP GTPases. Therefore, we propose PbCBP-O is not a homologue of the human guanine exchange factor, EPAC, which concurs with previous findings by Patel, Perrin and others for the *P. falciparum* homologue [9]. To gain insight into function of CBP-O, we utilised *P. berghei* parasites and deleted the complete ORF of the gene. Similar to observations in *P. falciparum* [9], we detected no impact on the asexual life cycle. It is interesting to note that the *cbp-o* gene has been reported to contain loss-of-function mutations in long-term culture adapted strains, such as *P. falciparum* 3D7 or HB3 [7]. Curiously, culturing of single or multiple genotype *P. falciparum* clinical isolates over several months also led to the emergence of loss-of-function mutations [8,35]. In light of our findings that PbCBP-O promotes parasite midgut invasion, it would be useful to assess the mutational status of PfCBP-O in culture adapted *P. falciparum* strains routinely used for mosquito transmission studies.

Notably, we show PbCBP-O is important for transmission into the mosquito and when the protein is absent, ookinetes reach maturity but fail to traverse the mosquito midgut epithelium effectively. Recent studies revealed that the conoid is conserved in *Plasmodium* [10,36]. Even when molecular composition of the conoid is maintained in different *Plasmodium* zoites, essentiality of components could be dependent on the different host environments and tissues encountered by the zoites. Accordingly, although CBP-O is expressed at the tip of merozoites and ookinetes, we observed invasion defects only in ookinetes. Ookinetes encounter a hostile gut environment and to ensure survival, they might require higher invasion competence to penetrate and cross the mosquito gut epithelium.

Our *in vitro* and *in vivo* studies demonstrated that the putative cyclic nucleotide binding domains of PbCBP-O are indeed functional and bind to both cAMP and cGMP. Interplay of cAMP and cGMP signalling in the related Apicomplexan, *T. gondi*, regulates parasite egress where PKA activity prevents premature egress of tachyzoites [37,38]. Moreover, TgPKA is required for shutting down PKG dependent tachyzoite motility to allow replication, when the parasite encounters the intracellular environment of the host cell. Similar to Toxoplasma, *P. falciparum* PKG regulates parasite egress [39]. In contrast, PfPKA activity does not influence merozoite egress and is essential specifically for erythrocyte invasion [9]. While there are lineage and developmental stage-specific differences, it is clear that cyclic nucleotide signalling is crucial for zoite motility and invasion. Intriguingly, PbCBP-O is expressed at the extreme apex of the ookinete tip and, in close proximity lies the ookinete extrados site (OES) which is the localisation site of GCβ or the cGMP synthesizer [40]. Considering that PbCBP-O localises close to the site of cGMP production, it is tempting to speculate that CBP-O regulates transduction of cGMP signals by either sequestering and/or sensing the cyclic nucleotide. Until now the only cGMP effector studied in *Plasmodium* was PKG and, in ookinetes the kinase promotes gliding motility [6]. High cGMP level is sustained in ookinetes and activity of the cGMP hydrolyser, PDEδ is presumably required for rounding up after the ookinete reaches the basal lamina of the midgut [4,5]. PbCBP-O mutant ookinetes show normal movement implying that requirement for CBP-O is subsequent to initiation of gliding motility (Fig 3A; S1 - S4 Movies). This suggests that spatial and temporal cGMP activity powers both ookinete motility and penetration of the mosquito via distinct transducers. In *Plasmodium* ookinetes, microneme secretion is controlled by

PKG to support gliding motility [41]. Microneme exocytosis in ookinetes also drives adhesion, and mosquito midgut traversal [22,42,43]. We did not observe defects in expression of several micronemal proteins, nor in chitinase secretion, a micronemal protein required for ookinetes to cross the peritrophic matrix (PM), the structure which encases the infected blood meal in the mosquito midgut [24,44]. Moreover CBP-O is not required for the asexual lifecycle, therefore a role in microneme expression or secretion is unlikely. However, micronemes show broad cargo diversity in the related Apicomplexan, *T. gondi* [45] and, further examination will determine if cohorts of ookinete-specific micronemes are CBP-O controlled.

Alternatively, cGMP action is restricted to regulating motility and cAMP signalling takes control during crossing of the epithelial cell barrier. The role of cAMP in ookinete biology remains unexplored. However, Adenyl cyclase β, a cAMP producer, and both the cAMP binding and catalytic subunit of protein kinase A (PKA-R and PKA-C) are expressed in mature ookinetes implying a functional cAMP signalling module in ookinetes [6,46]. The ability of CBP-O to bind cAMP reveals another putative player that could coordinate epithelial layer crossing by the ookinete in a cAMP dependent fashion. Further biochemical studies with isothermal titration calorimetry could determine the selectivity and specificity of CBP-O's CNB domains to cGMP or cAMP. Nevertheless, our data reveals a temporal nature of cyclic nucleotide signalling with discrete roles in ookinete motility and penetration of mosquito tissue.

While investigating the individual contribution of the N- and C-terminus, we discovered the N-terminus is required for positioning of CBP-O to the ookinete apex and, the C-terminus ensures binding to cyclic nucleotides. Previously published bioinformatic analysis of CNB domain containing proteins in eukaryotic pathogens identified amino acid variations in the nucleotide binding pocket and significant number of insertions within the CNB domain [47]. The authors hypothesized that the divergences could facilitate pathogen specific protein-protein interactions or alter nucleotide binding properties. Here we provide the first experimental evidence that, despite the amino acid variations and an insert of >20 amino acids, CNB3 domain of CBP-O retains the capacity to bind cyclic nucleotides (Fig 6C). Our findings will encourage further structural and biochemical studies to determine nucleotide specificity of the divergent CNB domains of CBP-O. Experiments should explore whether the divergence allows CBP-O to interact with cGMP and cAMP simultaneously or sequentially and if CBP-O regulates transition of cyclic nucleotide signalling from cGMP to cAMP. In ookinetes this relay could connect initiation of gliding motility to zoite invasion of the mosquito midgut epithelium. Moreover, it will be interesting to investigate if the *T. gondi* orthologue performs a similar role by linking cGMP and cAMP signalling to regulate tachyzoite motility and host cell invasion.

In conclusion, our findings expand the repertoire of functional cyclic nucleotide effectors in *Plasmodium* and demonstrate that PbCBP-O presents affinity for both cAMP and cGMP. Moreover, CBP-O conservation throughout Apicomplexa will provide opportunities to mechanistically understand how the zoite apex in these parasites utilise cyclic nucleotide signalling for host cell interaction and invasion.

## Materials and methods

### Ethics statement

All animal work was performed in accordance with the UK Animals (Scientific Procedures) Act 1986 (amended in 2012) and with European Directive 2010/63/EU on the Protection of Animals Used for Scientific Purposes. All procedures were approved by the University of Edinburgh's animal welfare and ethics board, and UK's Home Office (project license: PPL P04ABDCAA6). *P. berghei* parasites were maintained in CD-1 mice (Charles River) weighing between 22–32 g.

### Generation of targeting constructs, parasite transfection and diagnostic PCRs

Transgenic parasite lines were generated using the CRISPR-Cas9 system. For all PCR reactions Q5 polymerase (NEB) was used and plasmid assemblies were performed using the NEBuilder HiFi DNA Assembly (NEB) or T4 ligase (NEB).

Vector (named ENP119 here) expressing the gRNA from the PyU6 promoter, a cassette that contains the selectable markers *hdhfr* and *yfcu* and a humanised SpCas9 driven from the *Pbeef1*-α promoter was linearised with BsmBI to allow incorporation of the gRNA [48].

All homology arms ranged between 800 and 1200 base pairs. To generate PbΔCBP-O parasites the 5' homology arm was amplified using EU503 and EU504 and digested with HindIII and SacII. For the 3' homology arm primers EU492 and EU493 were used and digested with KpnI and NheI. These fragments were inserted sequentially into ENP119, to which the gRNA (EU564) was previously inserted. A *gfp* cassette flanked by *Pbeef1*-α promoter and *Pbcalmodulin 3'utr* was ligated into the vector using KpnI and SacII restriction enzyme sites. The plasmid (ENP142) was transfected into the HPtbb and 1804cl1 parental lines.

All domain mutants and GFP tagged CBP-O were created in 1804cl1 parental line. To generate CBP-O(ΔCNB4) parasites, primers EU704 and EU705 were employed to amplify the 5' homologous arm and, gfp was amplified with EU751 and EU752 from plasmid pG548. The previously constructed ENP142 vector was digested with HindIII and KpnI to remove the 5' homology arm and the GFP cassette used for generating ΔCBP-O lines, leaving only the 3' homology arm. The new 5' homology arm and *gfp* were then ligated into this vector. Subsequently, HindIII and EcoRI were employed to release the fragment containing the homology arms and the GFP tag which was ligated into ENP119 containing the gRNA primer EU701. To generate CBP-O(ΔCNB1–4/X), the 5' homologous arm was amplified with primers EU749 and EU750 from genomic DNA and the GFP tag was amplified with EU751 and EU752 from plasmid pG548 [49]. The gene targeting construct was assembled similarly to CBP-O(ΔCNB4) except for the gRNA (EU700). To generate CBP-O(ΔN-ter), primers EU884 and EU885 were used to amplify the 5' homology arm and the 3' homology region was amplified with primers EU888 and EU889, and the GFP tag was amplified from pG548 using EU886 and EU887. The pieces were ligated into vector ENP119 containing the gRNA primer EU883.

C-terminal GFP tagged CBP-O parasite lines were generated using the PL0031 vector using single crossover homologous recombination containing 1 kb of the *cbp-o* C-terminus amplified by using primers EU706 and EU707. The vector was linearised with BamHI prior to transfection.

Schizonts for transfection were obtained from culture, and selection of mutant parasites was performed as previously published [50]. Correct integration of the targeting plasmid and gene disruption/modification was verified by diagnostic PCRs with primers listed in S1 Table.

### *P. berghei* infections and phenotyping

Mice were administered phenylhydrazine (12.5 mg/L) by intraperitoneal injection two days prior to infection. Infections were established by injecting 150–300 μl of cryopreserved stocks. Mice used for transmission experiments were not administered phenylhydrazine. Parasitaemia was monitored by Giemsa staining of thin smears from a drop of tail blood. Parasites were observed under a standard light microscope with a 100X oil objective.

For generating mature schizonts, infected red blood cells were added to schizont media (RPMI with 25 mM HEPES, 20% FBS, 10 mM sodium bicarbonate, 0.38 mM hypoxanthine and 100 U/ml penicillin, 100 μg/ml streptomycin) and cultured over night at 37°C shaking at 40 RPM. For enrichment of gametocytes, mice were provided sulfadiazine (30 mg/L) in drinking water when parasitaemia reached between 1–6%. 48 hours later, gametocytes were retrieved through cardiac puncture. For ookinete cultures, gametocyte enriched blood was placed in ookinete media (RPMI with 25 mM HEPES, 20% FBS 4 mM sodium bicarbonate, 100 μM Xanthurenic acid, pH 7.8). Ookinetes were cultured for 24 hours at 21°C and mature ookinetes were assessed by Giemsa smears. Ookinete conversion rates were calculated from Giemsa staining or immunofluorescence using FITC-P25 antibody (see immunofluorescence assay methods). Conversion rates were determined by comparing mature ookinete (MO) numbers to total number of round (RO), retort (RE) and mature ookinetes $\left(\text{MO} = \left(\frac{\text{MO}}{\text{MO}+\text{RO}+\text{RE}}\right) * 100\right)$

To analyse exflagellation rates, blood obtained from cardiac puncture was immediately placed in exflagellation media (RPMI with 25mM HEPES, 4mM Sodium bicarbonate, 100uM XA, pH 7.8) at 21°C and 10 µl were placed in a haemocytometer. After 15 minutes, exflagellation centres, defined as cell clumps, were counted using a 10X objective with a light microscope.

## Transmissions and mosquito dissections

For transmissions, 100–300 female *Anopheles stephensi* mosquitoes were starved for 24 h and subsequently fed on anaesthetised, infected mice with a parasitaemia between 1–6%. Mosquitoes were dissected 6–14 days after transmissions to count oocyst burden in the mosquito midgut, or 19–21 days to remove the salivary glands. Fluorescent oocysts and salivary glands were observed using a Leica MZ 10F dissecting microscope, and images were taken with the Leica DFC3000 G camera and the Leica X software. Oocyst numbers were counted using ImageJ.

For sporozoite counts, salivary glands were dissected in 1X PBS 19–21 days after infection. Complete sets of salivary glands (2x3 lobes) from 5 mosquitoes were transferred to a microcentrifuge tube containing 100 µl of 1X PBS and were homogenized with a pestle for one minute. The microcentrifuge tube was spun for 3 minutes at 3500 g and the supernatant was removed leaving 15µl to resuspend the sporozoites. Sporozoite numbers were counted using a haemocytometer.

For transmission bite-backs, mosquitoes were allowed to feed on anaesthetised naive mice for 10 minutes between days 19–21 after their infection. Parasitaemia was monitored through Giemsa smears from day 3 until a parasitaemia of 3%, or if no parasites were observed twelve days after the bite, mice were culled.

For *in vivo* ookinete conversion assays, mosquitoes were infected with wild-type or ΔCBP-O parasites and 19 hours later midguts were isolated in 1X PBS. Midguts were cut lengthwise to remove the blood bolus which was transferred to a microcentrifuge tube. The solution was centrifuged at 1000 g for 2–3 minutes and the cell pellet was smeared in a glass slide. Ookinetes in the blood bolus were counted via Giemsa staining.

## Ookinete motility

Ookinetes were purified on a MACS LD magnetic column (Miltenyi). The column was pre-wet with enriched PBS (1X PBS supplemented with 20mM glucose, 4mM NaHCO$_3$ and 1% BSA). The ookinete culture was passed twice through a 27G needle to dissociate any ookinete clusters, and subsequently added to the column. The column was washed with 4 ml ePBS and ookinetes eluted by passing another 4ml of ePBS after removing the column from the magnet. An extra 2ml of ePBS were added and a plunger used to ensure all parasites were retrieved. Enriched ookinetes were resuspended 1:1 ratio with Matrigel. 10 µl of this solution were added to a glass slide and a coverslip was added and sealed with nail varnish. The samples were placed in the 21°C incubator for 30 minutes. Ookinetes were imaged in the Zeiss Imager.Z2 microscope fitted with the Prime BSI camera (Photometrics). The 40X objective was used to take sequential images every 20 seconds for 15 minutes and only ookinetes whose tracks were detectable for at least 5 minutes were considered for analysis. Images were processed with ImageJ using the MTrackJ plugin to retrieve speed values.

## Ookinete invasion assay

Mosquitos were fed on infected mice and 19–24 h later midguts were retrieved as described above (see transmissions and mosquito dissections). Midguts were either directly imaged under the microscope or by immunofluorescence assays. For immunofluorescence, after removal of blood bolus, midguts were fixed in 4% formaldehyde (Fisher Scientific) with 0.075% glutheraldehyde (Sigma) for 1 hour at room temperature. Then midguts were washed two times in 1X PBS and blocked in 3% BSA/PBS for 1 hour. Parasites embedded in the midgut were stained using FITC-P25 antibody incubated overnight at 4°C. The next morning, midguts were washed twice with 1X PBS for 15 minutes and incubated in 1X PBS supplemented with 10 µM Hoechst (Fisher Scientific) for 15 minutes. Excess Hoechst was removed with two washes of

1X PBS for 15 minutes. Midguts were then dried and flattened against the glass slides and mounted in clear Fluoromount G mount (Southern Biotech) overnight using a 22mm coverslip. Images were acquired with the Leica DM 2000 LED microscope, fitted with the Leica EL6000 external light source and a 20X objective. Images were taken with the Leica DFC3000 G camera using the Leica X software. Z-stacks were taken with a 20X objective on the Zeiss Axio Observer microscope fitted and Prime BSI camera (Photometrics) with a depth of 1µm and between 20–30 slices to cover the whole midgut. Images were analysed using ImageJ.

For direct imaging of infected midguts, mosquitoes were infected with RFP fluorescent parasites. After mosquito infection and midgut dissection, midguts were incubated in 1X PBS with 10 µM Hoechst for 15 minutes. Midguts were then washed twice with 1X PBS, and subsequently dried and flattened by creating a lengthwise cut using a needle and placing a 22mm coverslip. Midguts were imaged as described above.

### Immunofluorescence (IFAs) assays of schizonts and ookinetes and, live microscopy of ookinetes

For IFAs, 1ml of cultured ookinetes or mature schizonts were spun down at 2000g to pellet cells, the supernatant was removed, and the cell pellet was smeared on a glass slide. Smears were fixed in in 4% formaldehyde (Fisher Scientific) with 0.075% glutheraldehyde (Sigma) in PBS for 10 minutes, washed three times in 1X PBS and permeabilised with 0.1% Triton X-100/PBS for 10 minutes at room temperature. Slides were blood with 3% BSA/PBS for 1 hour at room temperature. The primary antibody was diluted to a desired concentration in 3% BSA/PBS and slides were incubated with it overnight at 4°C. Slides were then washed three times in 1X PBS for 10 minutes to remove excess primary antibody. Secondary antibody was diluted (1:2000) in 3% BSA/PBS and applied to slides and incubated for 1 hour and 30 minutes at room temperature. Slides were washed two times in 1X PBS for 15 minutes. The slides were mounted using 30µl of Fluoromount-G with DAPI (Fisher Scientific) and a 40mm coverslip and left overnight to set. Images were acquired using a 100X oil objective on the Leica DM 2000 LED microscope, fitted with the Leica EL6000 external light source and images were taken with the Leica DFC3000 G camera using the Leica X software. Images were analysed on ImageJ.

For live imaging of ookinetes, 10 µl of ookinete culture was pipetted onto a glass slide and subsequently covered with a 44mm coverslip. Ookinetes were imaged using a 100X oil objective on a Nikon Spinning Disk Confocal microscope fitted with a Prime BSI camera (Photometrics). Z-stacks of 15–20 slices were acquired with a depth of 0.3 µm. Images were analysed on ImageJ.

### Microneme proteins and secretion assay

For determination of micronemal protein expression, ookinete cultures of wild-type and ΔCBP-O parasites were spun down at 400g for 5 minutes and supernatant removed. Pellets of ookinetes generated from $10^6$ gametocytes were resuspended in erythrocyte lysis buffer (1.5M NH4Cl, 0.1M KHCO3, 0.01M EDTA) and incubated on ice for 20 minutes. The solutions were then centrifuged at 2800rpm at 4°C and washed with 20ml of cold PBS and centrifuged again to pellet the parasites. Pellets were lysed in RIPA buffer (10mM Tris-HCl pH 8.0, 140mM NaCl, 1mM EDTA, 1% Triton X-100, 0.1% sodium deoxycholate, 0.1% SDS, 1mM DTT) complemented with protease inhibitor cocktail (Complete Mini EDTA free tablets, Roche). Clarified lysates were resuspended in Laemmli sample buffer and separated by SDS-PAGE on a 4–20% Mini-PROTEAN TGX stain-free gel (Bio-Rad). Samples were then transferred onto nitrocellulose membranes (Amersham Bioscience) and proteins detected though immunoblotting using 1:6000 rabbit anti-CTRP, 1:6000 rabbit anti-chitinase, 1:4000 anti-WARP and 1:1000 rabbit anti-actin diluted in 5% milk/PBST [51]; and secondary antibodies (1:4000 goat anti-rabbit in 5% milk/PBST). Results were visualised with ECL kit (Amersham). For microneme secretion assay, ookinetes (from a 2% gametocytemia culture with 70% ookinete conversion rate) were purified and on a LD50 magnetic column and incubated in PBS for 2 hours [51]. The cells were spun down at 700×g and the supernatant collected, filtered through a 2 µm filter and concentrated on an Amicon-ultra centrifugal filter, 30K cut-off (Millipore), mixed with Laemmli sample buffer and Chitinase secretion was analysed by western blotting as described above.

## Ookinete injections into mosquito

Ookinetes were purified from *in vitro* cultures as described previously and resuspended in ookinete medium to achieve a concentration of 700 ookinetes/100 nl. The thorax of thirty *A. stephensi* mosquitoes (5–7 day old) were injected with 700 ookinetes each using glass capillary needles and the Nanojet III microinjector (Drummond Scientific). Salivary gland sporozoites were counted at 19dpi as described above.

## Quantitative real-time PCR

Four biological replicates of *A. stephensi* mosquito infections were performed with *P. berghei* 1804cl1 and ΔCBP-O (in 1804cl1 background) parasites and uninfected control. Blood-engorged midguts were dissected 22–24 hours post-blood feed. Twenty-five mosquito midguts were pooled for each group and stored in RNA*later* (Thermofisher) at -80°C. Total RNA was isolated from homogenised midguts using the miRVana isolation kit (Thermofisher), treated with TURBO DNAse (TURBO DNA-free, Thermofisher) and cDNA synthesised using SuperScript IV Reverse Transcriptase (Thermofisher) according to manufacturer's protocols. Transcript expression was quantified using Power SYBR green (Applied Biosystems) and gene specific primers (S1 Table) on a QuantStudio 5 Real-Time PCR System (Applied Biosystems). *SRPN6* gene expression was normalized against *Asrps7* using the ΔΔCt method.

## Recombinant protein expression

Coding sequences of PKG-D, CNB1,3 and 4 of PbCBP-O were amplified from *P. berghei* ANKA genomic DNA using primer pairs EU831/EU832 for PKG-D, EU839/EU840 for CNB1, EU835/EU836 for CNB3 and EU837/EU838 for CNB4 (S1 Table). PKG-D, CNB3 and 4 encoding DNA segments were cloned into Xho/NdeI site and CNB1 into XhoI/NheI of pET28a (Novagen) allowing expression of the CNB domains with a N-terminal 6X-histidine tag.

The pET28a:CNB1 was transfected into BL21-CodonPlus-RIL (Novagen) competent cells *E. coli* cells and pET28a:PKG-D, pET28a:CNB3 and pET28a:CNB4 were transfected into Rosetta DE3 (Novagen) competent cells. Cells were grown overnight in terrific broth (TB) at 37°C and then diluted 1:100 and grown in TB until an OD600 of around 0.7. CNB1, CNB3 and PKG-D expression was induced by adding IPTG at 0.5mM for 5 hours at 30°C. CNB4 expression was induced by adding 0.1mM IPTG overnight at 21°. Cells were spun at 4°C at 1800 rpm for 10 minutes to pellet the cells. The supernatant was removed and pellet stored at -80°C until further processing.

## Solubility tests were performed by lysing bacterial pellets in 1X BugBuster

(Novagen) complemented with protease inhibitors (complete Mini EDTA free tablets, Roche) and 0.5 µl benzonase (Sigma). Lysis was performed at room temperature for 30 minutes and then spun at 14,000 rpm for 15 minutes. Supernatant was mixed with 6x Laemmli sample buffer and run on an SDS-polyacrylamide gel. CNB1, CNB3 and PbPKG-D were detected in the soluble fraction while CNB4 was found in the insoluble fraction.

To recover CNB4 protein, the bacterial pellet was lysed with 2ml of 1X BugBuster and protease inhibitors and 0.2 µl of Benzonase by rotating at 30 rpm for 30 minutes at room temperature. The mixture was centrifuged for 10 minutes at maximum speed (30,000 g) at 4°C and supernatant removed. The pellet was washed twice in 1ml of 1X BugBuster and then spun at maximum (30,000 g) speed for 5 minutes at 4°C. To the final pellet 2 ml of buffer A were added (1xPBS, 8M Urea, 1mM DTT, protease inhibitor cocktail and 0.2 µl of benzonase), and mixed by vortexing for 1–2mins. The suspension was mixed by axial rotation (30 rpm) for 120 mins at 4°C. The suspension was centrifuged at 30,000 g for 15 mins and the supernatant was isolated. The sample was then dialysed to remove urea and allow refolding of the protein. The lysate was added to the dialysis cassette which was immersed in 500 ml of cold buffer B (1xPBS, 1M Urea, 1mM DTT) and dialysed for 2 hours at 4°C when the buffer was exchanged for fresh buffer and dialysed for another 2 hours. The dialysis cassette was then immersed into buffer C (1X PBS, 1mM DTT) and dialysed overnight at 4°C. The dialysed sample was mixed with glycerol to a final concentration of 20% and stored in the freezer.

## Cyclic nucleotide binding assay

**Recombinant proteins**: Soluble proteins were obtained by lysing bacterial pellets in 1.5 ml of buffer D (50 mM Hepes pH 7.4, 1 mM EDTA, 1% Nonidet P-40, 10 mM NaF, 0.1 mM sodium orthovanadate, 1 mM dithiothreitol) supplemented with protease inhibitor cocktail (Complete Mini EDTA free tablets, Roche). Samples were sonicated 3 times, 1.5 µl of Benzonase were added and sample were mixed by axial rotation at 4°C for 30 minutes. Samples were centrifuged for 10 minutes at 14,000 rpm at 4°C. CNB4 was obtained as described above and diluted to a 10% glycerol concentration. The cAMP and cGMP agarose gel matrix (8-AEA-cAMP-Agarose and 8-AET-cGMP-Agarose, BioLog) were diluted 1:1 in sterile PBS. For PKG-D, CNB1 and CNB3, matrix was washed in buffer D and for CNB4 gel was equilibrated in 1xPBS containing 10% of glycerol. For each assay 200 µl of the lysates or purified CNB4 were incubated with 100 µl of cAMP and cGMP agarose matrix for 2 hours and mixed by axial rotation (30 rpm) at 4°C. Matrix was spun down at 100 g for 30 seconds and the supernatant removed and kept as the flow through. Three washes were done with 1 ml of buffer D or 1xPBS (for CNB4) rotating for 5 minutes. For elution, the washed matrix was resuspended in 1x Laemmli sample buffer and heated for 15 minutes at 75°C. Elutes were separated on a 15% SDS-polyacrylamide gel and transferred to nitrocellulose membrane. Membrane was blocked in 5% skim milk in PBST buffer for 1h at room temperature and incubated with rat α-His antibody (Clonetech) at 1:6000 dilution for 1h at room temperature. Membrane was washed 3X15 min with PBST and incubated with HRP-conjugated α-rat IgG (CST) at 1:5000 dilution for 1h at room temperature. Membranes were washed three times in PBST followed by a final PBS wash and proteins visualized by ECL advanced kit (GE healthcare).

**Endogenous CBP-Os:** Pellets containing $1x\,10^7$ cells of GFP-tagged CBP-O(ΔN-ter) or CBP-O(ΔCNB1–4/X) expressing parasite lines from mature schizont stages with visible segmented merozoites were lysed in buffer D for 30 minutes on ice, and subsequently spun at 4°C for 15 minutes at 14,000 rpm. A small fraction of the lysate was resuspended in 6X Laemmli sample buffer to keep as the input sample, and the rest of the lysate was used for the binding assay. For this, 50 µl of previously PBS diluted cAMP and cGMP agarose gel matrix were washed in 200 µl buffer D. Parasite lysate was then incubated with the washed matrix for 2 hours at 4°C mixing by axial rotation (30 rpm). The beads were then spun down at 100 g for 1–2 minutes and supernatant was removed and kept as the flow-through sample. The matrix was washed with 500 µl buffer D on a rotator for 5 minutes for a total of three times. The washed matrix was resuspended in 1X Laemmli sample buffer and heated at 75 °C for 15 minutes. Bound elutes were separated by SDS-PAGE on a 4–20% Mini-PROTEAN TGX stain-free gel (Bio-Rad). Western blot analysis was performed as previously described using rabbit anti-GFP antibody (Chromotek) at 1:1000 dilution and HRP- conjugated anti-rabbit IgG (CST).

## AlphaFold structure predictions

The crystal structures of *Plasmodium falciparum* PKG-CNB-D or 4OFG [27] and PfPKA-R-CNB2 or 5KBF [30] were obtained from the RCSB Protein Data Bank (https://www.rcsb.org). The predicted structures of *P. berghei* CBP-O-CNB1–4 and CNBX were generated in the AlphaFold Protein Structure Database https://alphafold.ebi.ac.uk [52]. Structure predictions for *P. berghei* CBP-O-CNB1–4 and CNBX were generated using the ColabFold default MSA pipeline [53], monomer model and three cycles in the Google Colaboratory. The generated structures were annotated in PyMOL.

## Supporting information

**S1 Fig. Schematic representation of strategy to generate PbCBP-PO transgenic lines.** (A) CBP-O was C- terminally tagged with GFP employing a single crossover strategy where the gene was linearised using a BamHI site. (B) Full-length knock out was generated using CRISPR/Cas9 with a guide RNA mapping to the middle of the gene. The locus was replaced by a constitutively active GFP. Positions of oligonucleotides used for genotyping are indicated by arrows and identified by specific EU numbers. Genotyping by the respective diagnostic PCRs and expected sizes are indicated by agarose gel electrophoresis and a table respectively.
(TIF)

**S2 Fig. Schematic representation of strategy to generate PbCBP-PO mutant lines.** All mutant lines were created in constitutively expressing RFP line. (A) Full-length knock out (Δ*cbp-o*) was generated using the strategy as represented in SF1A. (B) *cbp-o(Δcnb1–4/x),* (C) *cbp-o(ΔN-term)* and (D) *cbp-o(Δcnb4)* transgenic lines were generated by replacing the indicated loci with *gfp* resulting in a GFP-tagged domain mutant. Positions of oligonucleotides used for genotyping are indicated by arrows and denoted by specific EU numbers. Genotyping by diagnostic PCRs and expected sizes are indicated.
(TIF)

**S3 Fig. Proportion of ookinetes displaying left-handed helical motion.** Ookinetes showing motility for at least 5 minutes were examined. n = 46 for each genotype and data is pooled from two independent replicates.
(TIF)

**S4 Fig. Contribution of N- and C-terminal domains on protein localisation.** (A) Deletion of CNB4 does not impact protein localisation to the apical tip of ookinetes. Immunofluorescence microscopy of mature ookinete expressing GFP-tagged CBP-O protein lacking the terminal CNB4 domain. (B) N-terminus is required for targeting CBP-O to the merozoite tip. The full-length and CNB domain mutant protein (ΔCNB1–4/X) maintain apical localisation. Immunofluorescence microscopy of mature schizonts expressing the various domain mutant proteins.
(TIF)

**S5 Fig. Recombinant protein expression of PbCBP-O's cyclic nucleotide binding domains.** Western blot analysis of His-tagged recombinant CNB1, CNB3 and CNB4 proteins. CNB1 and 3 were isolated from solution fractions after lysis under non-denaturing conditions. CNB4 was isolated from inclusion bodies in 8M Urea buffer, followed by refolding dialysis.
(TIF)

**S1 Movie. Motion of wildtype ookinetes (over 15 mins) in Matrigel tracked using MTrackJ plugin.**
(AVI)

**S2 Movie. Motion of PbΔCBPO ookinetes (over 15 mins) in Matrigel tracked using MTrackJ plugin.**
(AVI)

**S3 Movie. Magnification displaying left-handed helical motion of wildtype ookinetes.**
(AVI)

**S4 Movie. Magnification displaying left-handed helical motion of PbΔCBPO ookinetes.**
(AVI)

**S1 Table. Sequences of oligonucleotides used in the study.**
(DOCX)

**S2 Table. Values used to build graphs and statistical analysis.**
(XLSX)

## Acknowledgments

We thank the Centre Optical Instrumentation Laboratory (COIL) at the University of Edinburgh for training and use of their microscopes, Ronnie Mooney for providing mosquitoes, Dina Vlachou and Julia Cai (Imperial College) for helpful discussions and protocols for mosquito injections. We acknowledge Vector and Eukaryotic Pathogen Database (VEuPathDB) for their invaluable support for this study.

## Author contributions

**Conceptualization:** Dominika Kwecka, Nisha Philip.

**Formal analysis:** Dominika Kwecka, Nisha Philip.

**Funding acquisition:** Nisha Philip.

**Investigation:** Dominika Kwecka, Zhishuo Wang, Edvardas Eigminas, Lijia Liu, Choel Kim, Nisha Philip.

**Methodology:** Dominika Kwecka, Edvardas Eigminas, Lijia Liu, Jennifer C Regan, Choel Kim, Nisha Philip.

**Project administration:** Nisha Philip.

**Supervision:** Nisha Philip.

**Validation:** Dominika Kwecka, Nisha Philip.

**Visualization:** Dominika Kwecka.

**Writing – original draft:** Nisha Philip.

**Writing – review & editing:** Dominika Kwecka, Choel Kim.

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
