## [Decision Letter · Decision Letter 0]

27 Feb 2025

PPATHOGENS-D-25-00272

A divergent cyclic nucleotide binding protein promotes Plasmodium ookinete infection of the mosquito

PLOS Pathogens

Dear Dr. Philip,

Thank you for submitting your manuscript to PLOS Pathogens. After careful consideration, we feel that it has merit but does not fully meet PLOS Pathogens's publication criteria as it currently stands. Therefore, we invite you to submit a revised version of the manuscript that addresses the points raised during the review process.

Please submit your revised manuscript within 60 days Apr 28 2025 11:59PM. If you will need more time than this to complete your revisions, please reply to this message or contact the journal office at plospathogens@plos.org. Please include the following items when submitting your revised manuscript:

We look forward to receiving your revised manuscript.

Kind regards,

Mathieu Brochet

Academic Editor

PLOS Pathogens

Margaret Phillips

Section Editor

PLOS Pathogens

 Sumita Bhaduri-McIntosh

Editor-in-Chief

PLOS Pathogens

orcid.org/0000-0003-2946-9497

Michael Malim

Editor-in-Chief

PLOS Pathogens

orcid.org/0000-0002-7699-2064

**Additional Editor Comments :**

Three independent reviewers found your results to be exciting and believe they contribute significantly to our understanding of this signaling pathway in malaria parasites. However, several areas have been identified that require significant attention. Reviewers 2 and 3 have requested a more detailed phenotyping analysis of ookinete motility and midgut colonization. Please provide additional data and consolidate your findings to address their concerns. Reviewers 1 and 3 have raised concerns regarding the cNMP binding assay. They suggest determining the relative affinities to cAMP and cGMP, and, if appropriate, including additional replicates to strengthen your results. Additionally, please ensure that all other minor points raised by the reviewers are addressed either experimentally or within the text. We would also suggest discussing your results in light of the previously published roles of cAMP signaling following host cell invasion by Plasmodium merozoites or Toxoplasma tachyzoites.

**Journal Requirements:**

- TM on page: 23.

5) Thank you for stating that “All data will be available at journal website.” Please note that your Data Availability Statement is currently missing [the repository name and/or the DOI/accession number of each dataset OR a direct link to access each database]. Please provide a complete Data Availability Statement in the submission form, ensuring you include all necessary access information or a reason for why you are unable to make your data freely accessible. If your research concerns only data provided within your submission, please write "All data are in the manuscript and/or supporting information files" as your Data Availability Statement.

**Comments to the Authors:**

**Please note that one of the reviews is uploaded as an attachment.**

**Reviewers' Comments:**

Reviewer's Responses to Questions

**Part I - Summary**

Reviewer #1: This is a very interesting manuscript that reports on a Plasmodium berghei cyclic nucleotide-binding protein (PbCBP-O). The protein localises to the apex of ookinetes, a property that relies on the presence of its N-terminus. Disruption of the protein causes a markedly reduced ability of ookinetes to invade mosquitoes, even though ookinete gliding motility is not affected. Results were generated that are consistent with the conclusion that one of the predicted cyclic nucleotide-binding domains is able to bind to both cGMP and cAMP raising the intriguing possibility that PbCBP-O may be a mediator of both signalling pathways.

This study has been carried out carefully and the data represent an important contribution to our knowledge of cyclic nucleotide signalling in malaria parasites.

Reviewer #2: Please see the attached file containing my comments

Reviewer #3: Cyclic nucleotide signaling plays an essential role in several processes at multiple stages of Plasmodium. The parasite encodes 3 putative cNMP binding proteins, of which two (PKA and PKG) are well studied. There is a third protein with 4 predicted cNMP binding sites and a long N-terminal extension whose role is less well studied. It is known to be dispensable in the asexual erythrocytic stages.

The manuscript by Kwecka et al aims to fill this gap. They describe the essential functions of this protein, PbCBP-O in invasion by P. berhgei ookinetes of the midgut epithelium. Using parasites with GFP-tagged protein, they determine that it localizes to the apical end of ookinetes. They further characterize the protein by determining that 3 of 4 cNMP binding sites are essential for this function, while the amino end is required for proper localization to the apical end. Studies using recombinant individual cNMP binding sites were conducted to determine that each one is capable of binding both cAMP and cGMP. Using gene replacement, they demonstrate that parasites lacking the protein form ookinetes normally and these ookinetes appear to have normal motility and secretion of micronemal proteins. Intriguingly, they do not form oocysts. These results are interpreted to mean that ookinetes are unable to invade the midgut epithelium. The molecular genetic studies are well done and the manuscript is written clearly. The biochemical characterization and data on ookinete motility needs strengthening along with the assertion that CBP-O is an effector protein.

Major

Ookinete motility has various parameters – speed, displacement and chirality doi: 10.1111/cmi.12283. The authors should measure displacement and chirality.

Authors conclude that the lack of P25 staining implies lack of ookinete invasion. Alternative scenarios such as dysmorphic OES, increased ookinete nitrosylation, or clearance during epithelial migration should be examined. It would be advisable to use orthogonal P25-independent approaches to detect oocysts. Epithelial invasion by ookinetes causes upregulation of mosquito markers - their assessment can provide a direct assessment of ookinete invasive ability.

Figure 6: Knowledge of the binding affinities of each CNB site for cGMP and cAMP would be more relevant to determining the potential cNMP binding properties of the protein. Further, the sites may show cooperative binding as seen in PKG. It is surprising that CNB4, the most conserved site, did not show any binding, despite model predictions. Authors speculate that this results from improper folding. Have they tested different fusion proteins to improve solubility or CD studies to determine proper refolding?

The authors have the means to provide a more informative description of protein expression that is not limited to ookinetes. Line 286 states that the protein is expressed at the tip of merozoites but I did not find these data in the main text or in Supplementary information. Additionally, what is the protein’s expression and localization in sporozoites, the other motile and invasive stage in mosquitoes?

Discussion Lines 260-261 states that “cyclic nucleotide signaling through CBP-O is critical for P. berghei ookinetes to invade…”. I would urge caution in claiming that there is signaling through CBP-O or that CBP-O is an effector protein. Data presented in this manuscript, in my opinion, do not show an effect on signaling. There is a clear phenotype for sure, but is this due to impaired signaling? As noted by the authors, CBP-O does not appear to encode any identifiable domain except of CBPs, so is it a sensor or an effector?

Minor

Along the lines of the previous comment, the Discussion could include a Discussion of CBP-O’s potential role in sequestering cyclic nucleotides either to increase their local concentration or to prevent access by effector kinases.

**Part II – Major Issues: Key Experiments Required for Acceptance**

Reviewer #1: See below for suggested modifications to existing figures/text and comments.

Reviewer #2: (No Response)

Reviewer #3: Ookinete motility has various parameters – speed, displacement and chirality doi: 10.1111/cmi.12283. The authors should measure displacement and chirality.

Authors conclude that the lack of P25 staining implies lack of ookinete invasion. Alternative scenarios such as dysmorphic OES, increased ookinete nitrosylation, or clearance during epithelial migration should be examined. It would be advisable to use orthogonal P25-independent approaches to detect oocysts. Epithelial invasion by ookinetes causes upregulation of mosquito markers - their assessment can provide a direct assessment of ookinete invasive ability.

Fig 4D: The text claims that PbCBP-O binds cAMP and cGMP in vivo but results in Fig 4D do not support the assertion. It should include Pb-CBP-GFP parasites as well to provide context for the efficiency of cGMP binding in the deltaN mutant.

Figure 6: Knowledge of the binding affinities of each CNB site for cGMP and cAMP would be more relevant to determining the potential cNMP binding properties of the protein. Further, the sites may show cooperative binding as seen in PKG. It is surprising that CNB4, the most conserved site, did not show any binding, despite model predictions. Authors speculate that this results from improper folding. Have they tested different fusion proteins to improve solubility or CD studies to determine proper refolding?

The authors have the means to provide a more informative description of protein expression that is not limited to ookinetes. Line 286 states that the protein is expressed at the tip of merozoites but I did not find these data in the main text or in Supplementary information. Additionally, what is the protein’s expression and localization in sporozoites, the other motile and invasive stage in mosquitoes?

Discussion Lines 260-261 states that “cyclic nucleotide signaling through CBP-O is critical for P. berghei ookinetes to invade…”. I would urge caution in claiming that there is signaling through CBP-O or that CBP-O is an effector protein. Data presented in this manuscript, in my opinion, do not show an effect on signaling. There is a clear phenotype for sure, but is this due to impaired signaling? As noted by the authors, CBP-O does not appear to encode any identifiable domain except of CBPs, so is it a sensor or an effector?

**Part III – Minor Issues: Editorial and Data Presentation Modifications**

Reviewer #1: Line 43/44 Consider changing to ‘gametocytes are activated and develop into gametes’

Line 48 Consider ‘mediated’ rather than ‘organised’

Line 53 Consider ‘a strong motility defect’

Line 94 There is a 5th recognisable (highly degenerate) CNB site in the P. falciparum orthologue (between CNB3 and CNB4) though with a lower degree of conservation than the other four CNBs. I have not checked whether this is also the case in the P. berghei orthologue, but expect that it is likely present and if so this could be added to the narrative/figure.

Was salivary gland sporozoite load not determined for the CNB4 mutant (Figure 4F)? Presumably it would have been at ~WT levels.

An interesting finding that the N-term of the protein is required for apical localization.

I certainly think that e.g. the AlphaFold models of the cAMP-binding domains of P. falciparum PKAr should be included in Figure 6B given the conclusion that PbCBP-O can bind both cNMPs.

Line 216, consider replacing ‘organised’ by mediated.

Lines 206, 216, 217 and elsewhere should specify the ‘regulatory domain of PfPKA (PKAr)’

The data shown in Figure 6C-D are consistent with the conclusion that the protein can bind both cAMP and cGMP. Please add the number of biological and technical replicates that were carried out for these experiments to the Figure 6C and D figure legends (I may have missed the information). If they are single replicates the experiments should be repeated to increase confidence in this key result.

Line 268-269 This conclusion (that the protein is not an orthologue of mammalian EPAC) agrees with that drawn previously by Patel, Perrin et al 2019 regarding the P. falciparum orthologue of PbCBP-O (see Ref 9) and this should be mentioned.

It is interesting that PbCBP-O is localised conveniently close to GCβ. Could you get sufficient ookinete material for pull down experiments to explore potential binding partners in future work beyond the current manuscript?

It would definitely be good to explore in future work the suggestion that cAMP may interact with PbCBP-O during transit of the epithelial later since ACβ and PKAr are expressed in mature ookinetes. Is there any expression of PKAc in mature ookinetes?? It is intriguing that this protein may be a mediator of both cGMP and cAMP signalling in ookinetes (and perhaps blood stages).

Reviewer #2: (No Response)

Reviewer #3: Fig 4D is not mentioned in the text.

Materials should mention parasite stage for all experiments. Just one example, it is unclear if ookinetes or a different stage were used in Fig 4D.

PLOS authors have the option to publish the peer review history of their article (what does this mean?). If published, this will include your full peer review and any attached files.

Reviewer #1: No

Reviewer #2: No

Reviewer #3: No

**Figure resubmission:**
---

## [Decision Letter · Decision Letter 1]

18 Aug 2025

Dear Dr Philip,

We are pleased to inform you that your manuscript 'A divergent cyclic nucleotide binding protein promotes Plasmodium ookinete infection of the mosquito' has been provisionally accepted for publication in PLOS Pathogens, pending the incorporation of the minor revisions suggested by Reviewer 1..

Best regards,

Mathieu Brochet

Academic Editor

PLOS Pathogens

Margaret Phillips

Section Editor

PLOS Pathogens

Sumita Bhaduri-McIntosh

Editor-in-Chief

PLOS Pathogens

orcid.org/0000-0003-2946-9497

Michael Malim

Editor-in-Chief

PLOS Pathogens

orcid.org/0000-0002-7699-2064

Reviewer Comments (if any, and for reference):

Reviewer's Responses to Questions

**Part I - Summary**

Reviewer #1: The authors have addressed the points I raised.

Reviewer #2: In the revised manuscript, the authors have incorporated most of my comments and suggestions, and have convincingly addressed the issues raised by the two other reviewers. This is a well-conducted study.

Reviewer #3: The revision addresses concerns well through inclusion of new data and additional analysis.

**Part II – Major Issues: Key Experiments Required for Acceptance**

Reviewer #1: (No Response)

Reviewer #2: (No Response)

Reviewer #3: (No Response)

**Part III – Minor Issues: Editorial and Data Presentation Modifications**

Reviewer #1: Very minor suggestions:

Abstract

Line 14, change '..invasion of mosquito...' to e.g. 'invasion of mosquitos..'

Line 18, add a hyphen between 'nucleotide' and 'binding'

Author summary

Line 30, add 'the' between 'However' and 'functions'

Line 31, add a hyphen between 'nucleotide' and 'binding'

Line 35, add 'the' between 'both' and 'N-'

Reviewer #2: (No Response)

Reviewer #3: (No Response)

PLOS authors have the option to publish the peer review history of their article (what does this mean?). If published, this will include your full peer review and any attached files.

Reviewer #1: No

Reviewer #2: No

Reviewer #3: No

---

## [Editor Report · Acceptance letter]

Dear Dr Philip,

We are delighted to inform you that your manuscript, " 

A divergent cyclic nucleotide binding protein promotes Plasmodium ookinete infection of the mosquito," has been formally accepted for publication in PLOS Pathogens.

Best regards,

Sumita Bhaduri-McIntosh

Editor-in-Chief

PLOS Pathogens

orcid.org/0000-0003-2946-9497

Michael Malim

Editor-in-Chief

PLOS Pathogens

orcid.org/0000-0002-7699-2064